# Prolonging somatic cell proliferation through constitutive *hox* gene expression in *C. elegans*

Svenia D. Heinze[1,2], Simon Berger [1,3], Stefanie Engleitner[1,2], Michael Daube[1] & Alex Hajnal [1]✉

*hox* genes encode a conserved family of homeodomain transcription factors that are essential to determine the identity of body segments during embryogenesis and maintain adult somatic stem cells competent to regenerate organs. In contrast to higher organisms, somatic cells in *C. elegans* irreversibly exit the cell cycle after completing their cell lineage and the adult soma cannot regenerate. Here, we show that *hox* gene expression levels in *C. elegans* determine the temporal competence of somatic cells to proliferate. Down-regulation of the central *hox* gene *lin-39* in dividing vulval cells results in their premature cell cycle exit, whereas constitutive *lin-39* expression causes precocious Pn.p cell and sex myoblast divisions and prolongs the proliferative phase of the vulval cells past their normal point of arrest. Furthermore, ectopic expression of *hox* genes in the quiescent anchor cell re-activates the cell cycle and induces proliferation until young adulthood. Thus, constitutive expression of a single *hox* transcription factor is sufficient to prolong somatic cell proliferation beyond the restriction imposed by the cell lineage. The down-regulation of *hox* gene expression in most somatic cells at the end of larval development may be one cause for the absence of cell proliferation in adult *C. elegans*.

Most tissues and organs of higher organisms regenerate constantly throughout their lifetime. Regeneration depends on the function of adult somatic stem cells (SCs), pluripotent quiescent cells that can be activated during tissue homeostasis and repair throughout an organism's life[1,2]. The maintenance of an active SC population is tightly controlled to prevent pathologies. Impaired SC renewal causes defects in tissue repair and homeostasis, whereas uncontrolled renewal of SCs can lead to cancer formation[3]. Thus, the balance between SC proliferation and differentiation is maintained through interaction between the cell cycle machinery, chromatin remodeling factors and tissue-specific transcription factors[2,4]. In this context, much attention has been devoted to the role of *hox* genes, a conserved family of essential developmental regulators encoding

homeodomain transcription factors[5,6]. Different SC populations, such as hematopoietic, colonic and mesenchymal SCs rely on *hox* gene activity for their maintenance and regenerative capacity[7–9]. Besides their function in SC renewal and regeneration, de-regulated *hox* gene expression has been observed in different types of human cancer, such as acute myeloid lymphoid leukemia[10,11], breast[12] and colorectal carcinoma[7,13]. The oncogenic activity of *hox* genes may in part be due to their ability to directly stimulate cell cycle progression and DNA replication[14].

In contrast to higher organisms, somatic cells in *C. elegans* divide according to a fixed lineage pattern and irreversibly exit the cell cycle at the end of larval development[15,16]. Hence, no proliferation of somatic cells occurs in adult animals. The *C. elegans* genome encodes six *hox*

[1]Department of Molecular Life Sciences, University Zürich, Winterthurerstrasse 190, 8057 Zürich, Switzerland. [2]Molecular Life Science PhD Program, University and ETH Zürich, CH-8057 Zürich, Switzerland. [3]Institute for Chemical- and Bioengineering, ETH Zürich, Vladimir Prelog Weg 1, 8093 Zürich, Switzerland. ✉e-mail: alex.hajnal@mls.uzh.ch

genes, the anterior *ceh-13* (*lab*), the central *lin-39* (*Scr*) and *mab-5* (*Antp*) and the posterior group genes *egl-5* (*AbdB*), *php-3* and *nob-1*[6,17]. Inspection of developmental mRNA expression profiling data from the modEncode project[18] indicates that *C. elegans hox* genes are highly expressed during embryonic and early larval development, but their mRNA levels globally decrease during later larval development and reach their lowest levels in adults (Supplementary Fig. 1A). The *hox* co-factor *unc-62* (Homothorax)[19], on the other hand, remains expressed at comparably higher levels in adults.

The central *hox* gene *lin-39* plays multiple roles during the development of the vulva, the hermaphrodite egg-laying organ[20]. During the first larval stage (L1), *lin-39* expression specifies six out of twelve epidermal Pn.p cells in the mid-body region to form an equivalence group consisting of the vulval precursor cells P3.p to P8.p (VPCs). *lin-39* maintains the VPCs as polarized epidermal cells arrested in the G1 phase of the cell cycle, while the remaining five Pn.p cells that are outside of the equivalence group fuse with hyp7[21,22]. At the end of the L2 and beginning of the L3 larval stage, *lin-39* is involved in the specification of the primary (1°) and secondary (2°) vulval cell fates[20]. Once specified, the VPCs undergo three rounds of cell divisions generating 22 terminally differentiated vulval cells. *lin-39* was previously shown to be necessary for VPC cell cycle progression by activating the expression of core cell cycle regulators, such as the cyclin E homolog *cye-1* or the cyclin-dependent kinase *cdk-4*[22,23]. VPCs lacking *lin-39* retain the ability to form a vulval invagination, but cannot divide. *lin-39* remains highly expressed in the proliferating vulval cells until they have completed their last round of cell divisions in early L4[24] (and this study).

Here, we have investigated the role of *hox* genes in determining the temporal competence of different cell types to proliferate. Constitutive *lin-39* over-expression in the VPCs or the sex myoblasts (SMs) was sufficient to induce precocious cell division and prolong their normal proliferation period beyond their normal point of arrest. Furthermore, ectopic expression of different *hox* genes in the quiescent anchor cell (AC) caused the AC to re-enter the cell cycle during the L3 or L4 stage and proliferate until young adulthood. Our results indicate that *hox* expression levels are essential to maintain the proliferative capacity of somatic cells by activating the core cell cycle machinery. We propose that the rapid down-regulation of *hox* genes in most somatic cells at the end of larval development is one cause for the absence of cell divisions and regeneration in the adult *C. elegans* soma.

## Results

### LIN-39 dosage determines the duration of vulval cell proliferation

Previous studies have shown that *lin-39* is necessary for the division of the six VPCs[22,23,25], but it has been unknown if *lin-39* is also necessary to sustain the proliferation of the differentiating vulval cells after fate specification. To investigate this question, we first examined the expression pattern of an endogenous *lin-39* reporter carrying a C-terminal *gfp* insertion along with a *zf1* degradation tag (*lin-39[zh120[lin-39::zf1::gfp]*, this study) by live-imaging the dividing vulval cells from the late L2 until the mid-L4 stage using microfluidic devices[26] (Fig. 1A and Supplementary Movie S1). LIN-39::ZF1::GFP was dynamically expressed in the induced VPCs and their descendants until the third round of cell divisions in early L4, with the highest levels observed in the 1° lineage. Thereafter, LIN-39::ZF1::GFP expression rapidly faded in all vulval cells except for the ventral VulA cells, where expression persisted until the end of the L4 stage. Possibly, LIN-39 plays a role during the morphogenesis of the VulA toroids after the vulval cell divisions have been completed.

If *lin-39* is necessary to maintain cell proliferation, then depletion of LIN-39 after vulval induction should lead to premature termination of the vulval cell lineage. To test this hypothesis, we induced a conditional knock-down of *lin-39* at the late L2/early L3 stage after the VPC

fates have been specified but before the first round of cell division took place. For this purpose, we expressed the E3 ubiquitin ligase ZIF-1 under control of the heat-shock inducible *hsp-16* promoter (*hsp>zif-1*) in the *lin-39::zf1::gfp* strain[27]. We included an *ajm-1::gfp* reporter (*swIs79*) to label adherens junctions in the VPCs and their descendants[28]. Since a complete loss of *lin-39* function causes the VPCs to fuse with the surrounding hyp7 syncytium during the L1 stage, the AJM-1::GFP marker allowed us to distinguish fused from unfused VPCs, which have maintained their apicobasal polarity[22].

Heat-shock induction of *zif-1* at the mid-L2 stage resulted in LIN-39::ZF1::GFP degradation from the L3 stage onwards (Supplementary Fig. 1B). To further reduce LIN-39 levels, we performed RNA interference (RNAi) by placing the heat-shocked L2 larvae on bacteria expressing *lin-39* dsRNA (*lin-39i*) or on empty vector (*ev*) bacteria as negative control (Fig. S1B) and counted the numbers of differentiated vulval cells at the L4 stage (Fig. 1B, C and Supplementary Fig. 1C). Most control animals lacking the *zif-1* transgene (grown on *ev* bacteria and exposed to a heat shock) contained 22 vulval cells (*n* = 21), except for two cases with 21 and 20 cells, which could be due to the heat shock or to endogenous *zif-1* expression. In contrast, animals subjected to *zif-1* and *lin-39i*-mediated double knockdown contained on average 10.5 ± 4.4 SD (standard deviation) (*n* = 28) differentiated vulval cells, while *zif-1* or *lin-39i*-mediated single knockdown resulted in a weaker reduction in vulval cell numbers (Fig. 1C). As illustrated for the examples shown in Fig. 1B and Supplementary Fig. 1C, vulval cells lacking LIN-39 often failed to complete all three rounds of cell division. The remaining vulval cells that formed an invagination continued to express the AJM-1::GFP adherens junction marker, indicating that the cells had not fused with hyp7. Possibly, the down-regulation of *lin-39* was not strong enough to cause a loss of cell polarity, or *lin-39* may not be required to maintain the polarity of the vulval cells after they have been induced.

Taken together, the conditional knock-down experiments suggest that LIN-39 is necessary to maintain the proliferative capacity of the vulval cells.

### *lin-39* over-expression triggers early Pn.p cell divisions

We next tested if increasing the *lin-39* dosage was sufficient to induce additional cell divisions. For this purpose, we generated a transgene consisting of a genomic fragment spanning the five *lin-39* exons fused C-terminally to a *gfp* reporter and expressed under control of a 3.25 kb (kilo base pairs) *egl-17* enhancer/promoter fragment (*egl-17p>lin-39::gfp*) to over-express *lin-39* in the vulval cells (Fig. 2A). In early L2 larvae before vulval induction, the *egl-17* promoter has weak basal activity in all VPCs[29,30]. During vulval induction in late L2, *egl-17* expression is up-regulated in the 1° VPC (P6.p) and remains expressed in 1° vulval cells until the P6.pxx stage, after which expression vanishes in the 1° cells and the 2° VulC and VulD cells start expressing *egl-17*. The *egl-17p>lin-39::gfp* transgene was stably integrated into the genome (*zhIs167*) and combined with the *he317[eft-3p>LoxP-egl-13nls::bfp-LoxP-egl-13nls::mCherry]; heSi220[lin-31p>cre]* marker (*VPC>mCherry*) to label all Pn.p cells and their descendants until they fuse with hyp7[31]. As a negative control, we used an *nls::gfp* reporter (*nls* is a nuclear localization signal) driven by the same *egl-17* enhancer/promoter fragment on an extra-chromosomal array (*zhEx670[egl-17p>nls::gfp]*), combined with the *VPC>mCherry* marker (Fig. 2A).

*egl-17p>nls::gfp* control animals at mid L2 stage contained eleven Pn.p cells (P1.p - P11.p) as in the wild-type (Fig. 2A,A'). Note that at this point the anterior P1.p to P3.p and posterior P9.p to P11.p cells, which are not part of the vulval equivalence group, had already fused with hyp7 and showed only weak expression of the VPC>mCherry marker. Only very faint *nls::gfp* expression could be observed in the VPCs at this stage (not visible in Fig. 2A). The *egl-17p>lin-39::gfp* transgene, on the other hand, caused precocious divisions of the Pn.p cells in late L1/early L2 larvae, resulting in the formation of extra Pn.p cells in 80%

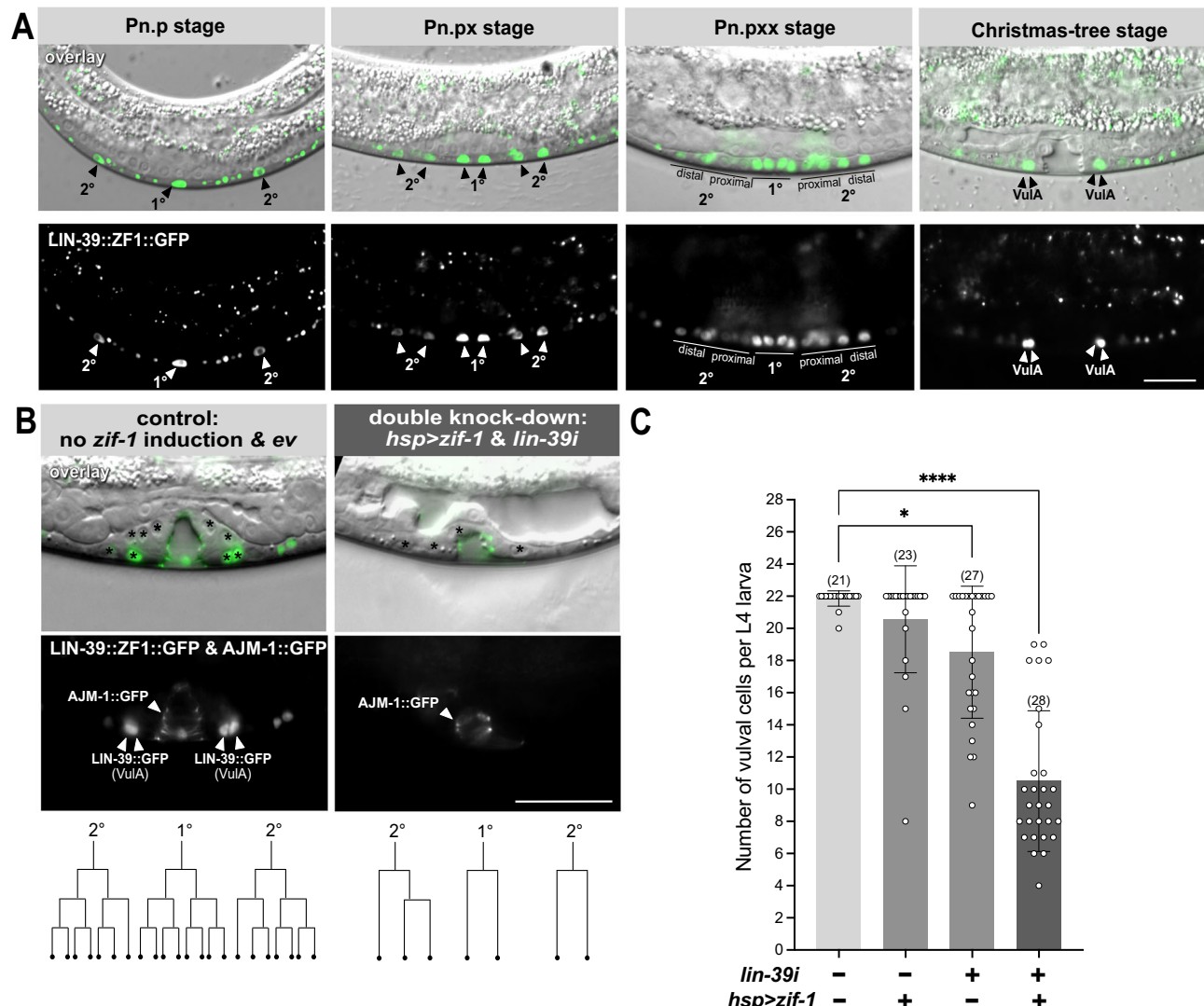

**Fig. 1 | Endogenous *lin-39::zf1::gfp* expression pattern and conditional *lin-39* depletion. (A)** Expression pattern of the endogenous *zh120(lin-39::zf1::gfp)* reporter at different stages of vulval development (see Supplementary Movie S1). Mid-sagittal DIC images overlaid with LIN-39::ZF1::GFP (green), with the separate GFP channel below. Animals were between the Pn.p (early L3) and the "Christmas tree" (L4.4) stage[32] after the completion of the vulval cell divisions. The 1° and 2° vulval cells are indicated with arrowheads or lines. Arrowheads in the L4.4 animal indicate the VulA cells where LIN-39::ZF1::GFP expression persisted until late L4. At least 20 animals were examined for each of the stages shown. **(B)** Conditional *lin-39* depletion. Left: Control *lin-39(zh120); swIs79[ajm-1::gfp]* L4 larva not carrying the *[hs-16p::zif-1]* array but subjected to a heat-shock in late L2 and grown on control RNAi plates. Right: *lin-39(zh120); swIs79[ajm-1::gfp]; zh616[hs-16p::zif-1]* L4 larvae subjected to an L2 heat shock and grown on *lin-39* dsRNA bacteria, as illustrated in Supplementary Fig. 1B. Mid-sagittal DIC images overlaid with the LIN-39::ZF1::GFP and AJM-1::GFP (green), with the separate GFP channel below. VPC nuclei in the DIC images are marked by asterisks, VulA cells and cell junctions with yellow arrowheads (see Supplementary Fig. 1C for individual z-sections). The corresponding cell lineages are indicated underneath. **(C)** Quantification of vulval cell proliferation after conditional *lin-39* depletion. All conditions included an L2 heat shock. Bars show the average cell numbers, error bars the standard deviations, and dots the individual values for the four indicated conditions. Two independent *[hs-16p::zif-1]* arrays (*zhEx 616.1&616.2*) were used. The numbers of animals scored are indicated in brackets. Statistical significance was determined with a Kruskal-Wallis non-parametric test followed by Dunn's multiple comparison correction and is indicated with **** for *p* = 1e-11 and * for *p* < 0.0382. Scale bars are 20 μm. Source data are provided as a Source Data file.

of the animals (*n* = 22), with an average of 13.3 ± 2.1 SD (*n* = 22) and a maximum of 19 Pn.p cells per animal (Fig. 2A, A'). To exclude the possibility that the Pn.p cell duplication phenotype in *egl-17p>lin-39::gfp* animals might be due to the chromosomal integration site of the *zhIs167* transgene, we examined an extra-chromosomal *egl-17p>lin-39::gfp* transgene and found a similar early Pn.p cell division phenotype (Supplementary Fig. 2A).

Furthermore, we observed enhanced and ectopic *egl-17p>lin-39::gfp* expression in the duplicated VPCs of L2 larvae, at a stage when *egl-17p>nls::gfp* expression in control animals was barely detectable. We hypothesized that the *egl-17p>lin-39::gfp* transgene may have created a positive feedback loop, since *egl-17* is likely a direct *lin-39* target

containing a *hox/pbx* binding site in a distal enhancer 1.58 kb upstream of the translational start site[29], and the first intron of *lin-39* contains another *hox/pbx* binding site[24]. Thus, the initially low *lin-39* levels driven by basal *egl-17* promoter activity may be self-amplified before vulval induction and maintained during the subsequent stages. Consistent with this hypothesis, expression of a full-length *lin-39* cDNA lacking the first intron did not induce early Pn.p divisions, while expression of a transgene containing a fragment of the first intron re-inserted upstream of *egl-17p>lin-39* cDNA restored the early Pn.p cell proliferation phenotype (Supplementary Fig. 2B-B").

Thus, increasing the LIN-39 dosage in the Pn.p cells resulted in precocious divisions during the late L1 or early L2 stage.

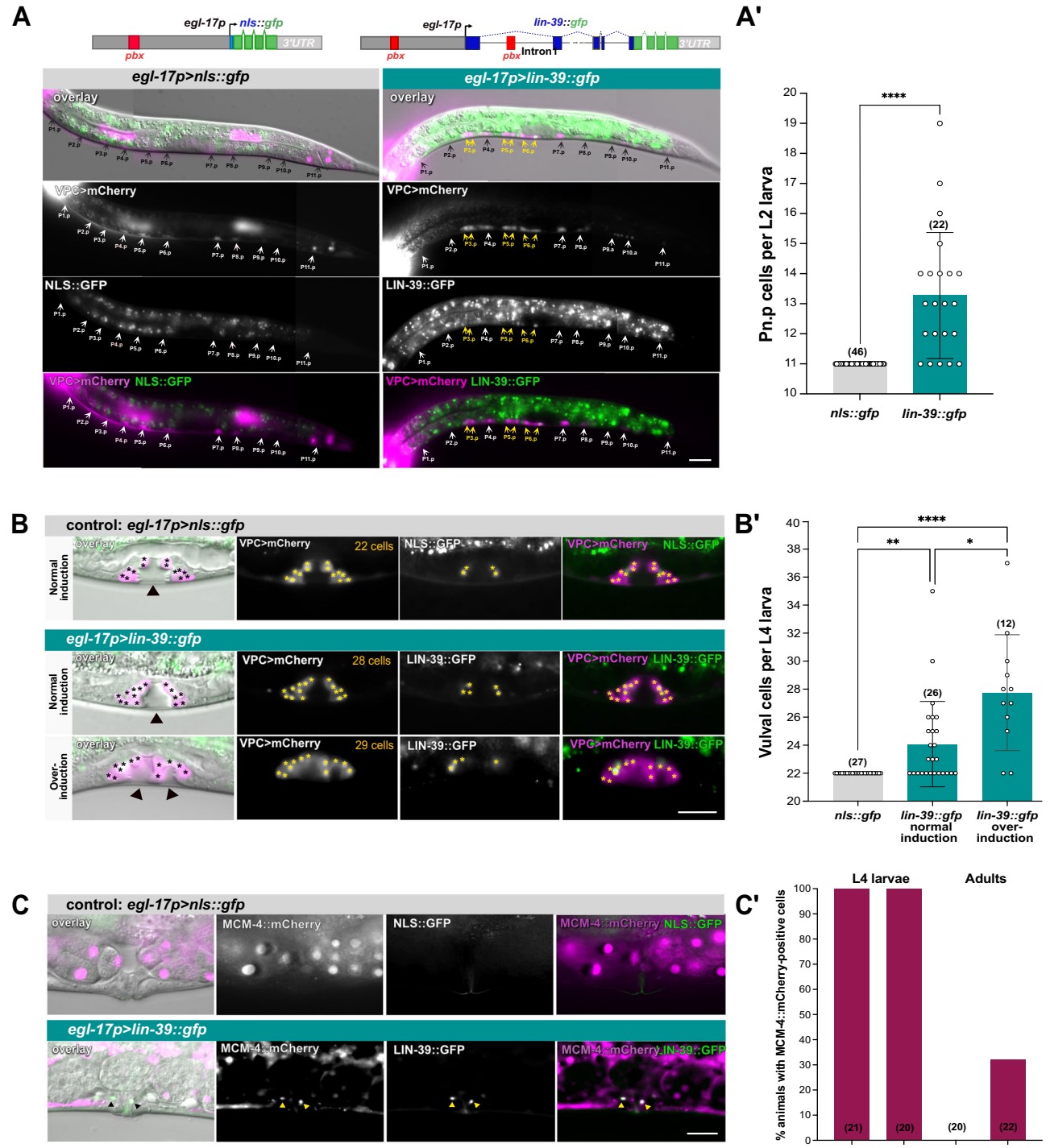

## Constitutive *lin-39* expression prolongs the proliferation of vulval cells

A second phenotype resulting from increased *lin-39* expression was an over-proliferation of the differentiating vulval cells during the mid to late L4 stage. While all *egl-17p>nls:gfp* control larvae contained 22 differentiated vulval cells at the late L4 stage, *egl-17p>lin-39::gfp* larvae often contained more than 22 vulval cells (Fig. 2B, B'). The late over-proliferation phenotype in *egl-17p>lin-39::gfp* larvae might be due to the ectopic induction of extra Pn.p cells generated during L2, or one or several of the proximal VPCs (P5.p-P7.p) that are normally induced could have undergone an extra round of cell division. To distinguish between these two scenarios, we performed live-imaging of *egl-17p>lin-39::gfp; VPC>mCherry* larvae from late L1 until early L4

(Supplementary Fig. 3A, B, Supplementary Movies S2 & S3). In 4 out of 7 animals, we observed at least one precocious Pn.p cell division during L2. Early divisions of distal cells, P3.p, P4.p or P8.p, did not affect vulval induction, since these cells adopted a 3° fate and fused with hyp7 at the Pn.px stage, as indicated by the disappearance of the VPC>mCherry signal (e.g. the P8.p daughters in Supplementary Fig. 3A, B). In 2 out of 4 animals, where one of the proximal Pn.p cells (P5.p, P6.p or P7.p) had duplicated during L2, four cells were induced to adopt 1° or 2° vulval fates, resulting in the formation of a second vulval invagination (over-induction in Fig. 2B and Supplementary Fig. 3B). In the remaining 2 animals, three Pn.p cells were induced to adopt vulval fates as in the wild-type, the extra Pn.p cell adopted a 3° uninduced cell fate and a single invagination formed (normal induction in Fig. 2B and

**Fig. 2 | *lin-39* over-expression induces precocious Pn.p and prolonged vulval cell division. (A)** Schematic drawings of the transgenes used (top). Images of L2 larvae with early Pn.p cell divisions in *zhIs167[egl-17p>lin-39::gfp]* (right side) compared to *zhEx670.1[egl-17p>nls::gfp]* controls (left side). Mid-sagittal DIC images overlaid with the NLS::GFP or LIN-39::GFP signal (green) and VPC>mCherry (magenta), with maximum intensity projections of the separate and merged fluorescence channels shown below. Black and white arrows indicate the single and yellow arrows the duplicated Pn.p cell nuclei. **(A′)** Quantification of Pn.p cell number in L2 larvae. Bars show the average, error bars the standard deviation and dots individual values. Three independent *[egl-17p>nls::gfp]* arrays (*zhEx670.1/ 670.2/670.3*) were used as controls. Statistical significance was determined with a two-tailed Mann-Whitney non-parametric test and is indicated with **** for *p* = 5.8e-12. **(B)** Late over-proliferation phenotype in *zhIs167[egl-17p>lin-39::gfp]* L4 larvae. The top row shows an *egl-17p>nls::gfp* control L4 larva with 22 vulval cells, the middle row an *egl-17p>lin-39::gfp* L4 larva with normal induction of the three proximal VPCs but 28 vulval cells, and the bottom row an *egl-17p>lin-39::gfp* L4 larva with an over-induction phenotype (4 VPCs induced) and 29 vulval cells. Mid-sagittal DIC images overlaid with the NLS::GFP or LIN-39::GFP (green) and VPC>mCherry (magenta), along with maximum intensity projections of the separate and merged

fluorescence channels are shown (see Supplementary Fig. 3C for individual z-sections). Vulval cell nuclei are labeled with asterisks. **(B′)** Quantification of the vulval cell numbers in L4 larvae. Bars show the average, error bars the standard deviation and dots individual values. Animals with normal and over-induction are shown separately. Statistical significance was determined with a Kruskal-Wallis non-parametric test followed by Dunn's multiple comparison correction and is indicated with **** for *p* = 1.7e-7, ** for *p* = 0,0013 and * for *p* = 0.0258. **(C)** MCM-4::mCherry expression in adult *zhEx670.1[egl-17p>nls::gfp]* control and *zhIs167[egl-17p>lin-39::gfp]* animals. Mid-sagittal DIC images overlaid with NLS::GFP or LIN-39::GFP (green) and MCM-4::mCherry (magenta), along with maximum intensity projections of the separate and merged fluorescence channels are shown. Black and yellow arrowheads indicate LIN-39::GFP and MCM-4::mCherry double-positive nuclei. **(C′)** Percentages of cells co-expressing NLS::GFP and MCM-4::mCherry in controls or LIN-39::GFP and MCM-4::mCherry in *egl-17p>lin-39::gfp* animals at the L4 stage and in 1 to 2 day-old adults. Two independent *egl-17p>nls::gfp* arrays (*zhEx670.1&670.2*) were used as controls. The numbers of animals scored are indicated in brackets. Scale bars are 20 μm. Source data are provided as a Source Data file.

Supplementary Fig. 3A). The observation that duplication of proximal but not of distal Pn.p cells occasionally resulted in an over-induced phenotype suggests that Pn.p cells expressing high levels of *lin-39* are more responsive to the inductive AC signal, but *lin-39* over-expression alone is not sufficient to induce vulval differentiation. Thus, an excess number of differentiated vulval cells in animals with a single vulval invagination at the L4 stage is indicative of extra vulval cell divisions rather than VPC over-induction. *egl-17p>lin-39::gfp* larvae with a single vulval invagination contained on average 24.1 ± 3.0 SD (*n* = 26) and up to 35 vulval cells, while animals with two invaginations contained on average 27.8 ± 4.2 SD (*n* = 12) and up to 37 vulval cells (Fig. 2B′ and Supplementary Fig. 3C). The presence of more than 22 vulval cells in animals with normal vulval induction suggested that the differentiating vulval cells had undergone one or more extra rounds of cell divisions, either by dividing faster during the normal proliferation phase or by dividing for an extended period during late L4. To distinguish between these two possibilities, we performed live-imaging of *egl-17p>lin-39::gfp; VPC>mCherry* larvae from the early L4 stage until young adulthood and counted the numbers of vulval cells formed per invagination at two different time points, first at the early L4 stage (L4.1 to L4.2 sub-stages) when all vulval cells have been formed in the wild-type[32] and then 6 to 9 hours later during the late L4 stage (L4.7 to L4.8) (Supplementary Fig. 3D, E). In 5 of the 21 animals imaged, extra vulval cells were already present at the early L4 stage, and their numbers did not increase later during L4. However, in 6 of the 21 animals vulval cell numbers further increased in late L4 larvae (highlighted in bold in Supplementary Fig. 3E; the remaining 10 animals exhibited no over-proliferation). Thus, over-proliferation not only occurred during the normal period of vulval cell divisions until early L4 but the proliferative phase was also prolonged at least until the late L4 stage.

Counting vulval cells in adult animals was not possible as the VPC>mCherry reporter started to accumulate in the cytoplasm, making it impossible to distinguish individual cells. We therefore monitored the expression of *mcm-4*, which encodes a subunit of the pre-Replication Complex (pre-RC)[33]. An *mcm-4::mCherry* reporter (*heSi4*) was expressed in all dividing cells including the vulval cells, as well as in G1-arrested cells that had not yet completed their lineage (e.g. the VPCs), but not in terminally differentiated quiescent cells in adults (Fig. 2C and Supplementary Movie S2). One day-old adult *egl-17p>lin-39::gfp* animals contained between two and eight LIN-39::GFP-positive cells, while no NLS::GFP expression could be detected in adult *egl-17p>nls::gfp* controls (Fig. 2C). Moreover, 32% (*n* = 22) of adult *egl-17p>lin-39::gfp* animals contained MCM-4::mCherry, LIN-39::GFP double-positive cells, while none (*n* = 20) of the *egl-17p>nls::gfp* adults showed any MCM-4::mCherry expression in the adult vulva (Fig. 2C, C′).

Taken together, constitutive *lin-39* over-expression resulted in an over-proliferation of vulval cells during the L4 stage. The persisting MCM-4::mCherry expression in young adults suggests that LIN-39::GFP-positive vulval cells may have maintained their proliferative potential during adulthood.

### *lin-39* over-expression induces excess sex myoblast proliferation

To further investigate the capacity of *lin-39* in stimulating cell proliferation, we assessed the consequences of *lin-39* over-expression in the sex myoblasts (SMs). Two SMs are born during the L1 stage in the posterior body region. During the L2 stage, they migrate to the mid-body region, where they divide three times during the L3 and L4 stages, producing eight vulval and eight uterine muscle cells each[16,34]. LIN-39 is expressed in the SMs and their descendants until they have terminally differentiated during the L4 stage, where LIN-39 activates different targets required for SM differentiation and proliferation[24,35].

We first followed endogenous LIN-39::ZF1::GFP expression in the SMs and their descendants from the late L2 until the L4 stage. LIN-39::ZF1::GFP was expressed in the two SMs in the mid-body region before their divisions and in all their descendants during the following divisions in the left and right body regions (Fig. 3A and Supplementary Movie S1). In contrast to the VPCs, LIN-39::ZF1::GFP expression levels in the SMs and their descendants were uniform. After the third round of cell divisions, eight LIN-39-positive SM descendants were aligned on each body side (Fig. 3A, bottom right panels). Similar to the vulval cells, LIN-39::ZF1::GFP expression faded after the last round of divisions and disappeared in adults.

Since *lin-39* and *mab-5* are redundantly required for SM proliferation[34], we tested if *lin-39* over-expression alone is sufficient to cause SM over-proliferation. To this end, we expressed a *lin-39::mCherry* minigene containing the first intron under the control of a 2.58 kb *hlh-8* promoter/enhancer fragment. *hlh-8* is expressed in all cells of the M lineage and activated by *lin-39* and *mab-5* in the SMs[34,36]. To create a positive feedback loop similar to the one in the VPCs expressing the *egl-17p>lin-39::gfp* transgene, we inserted a 161 bp fragment with the distal *egl-17* enhancer containing a *hox/pbx* binding site at the 5′ end of the *hlh-8* promoter/enhancer (*pbx-hlh-8p>lin-39::mCherry*, Fig. 3B). Animals carrying the *pbx-hlh-8p>lin-39::mCherry* transgene on an extra-chromosomal array (*zhEx678*) were scored at the mid to late L4 stage and compared to control animals carrying a *pbx-hlh-8p>nls::mCherry* transgene (*zhEx690*). Both strains in addition contained the *ayIs7[hlh-8p>gfp]* reporter (*SM>gfp*) to label the SM descendants with GFP[37]. In control animals, 16 cells (8 on each side) co-

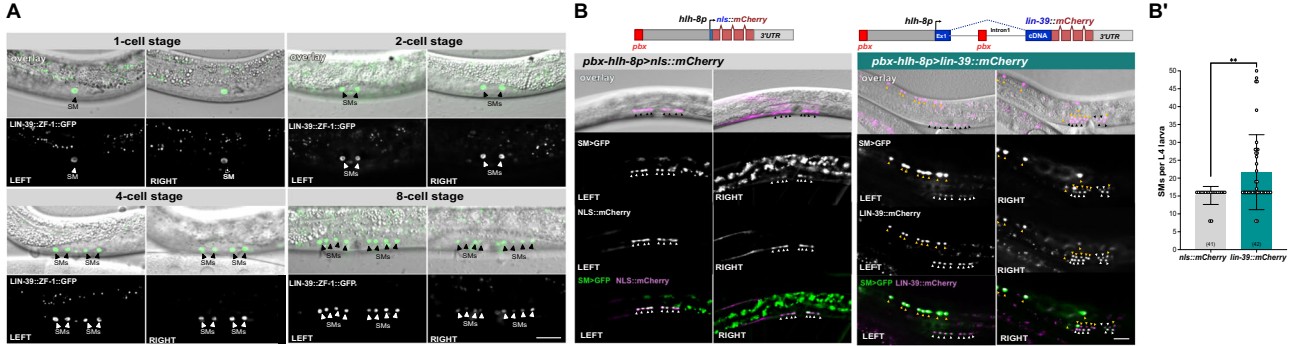

**Fig. 3 | *lin-39* over-expression induces excess SM proliferation. (A)** Expression pattern of the endogenous *zh120(lin-39::zf1::gfp)* reporter in the SMs at different stages of larval development (see Supplementary Movie S1). DIC images overlaid with the LIN-39::ZF1::GFP (green), with the separate GFP channel below. Animals were between the 1-cell (early L3) and the 8-cell stage (L4.4) after completion of the SM divisions. For each stage, z-sections of the left and right body sides are shown. Black and yellow arrowheads indicate the SM nuclei. At least 15 animals were examined for each of the stages shown. **(B)** Schematic drawings of the transgenes used for SM expression (top). Images show the SMs in *zhEx678[pbx-hlh-8p>lin-39::mCherry]* and *zhEx690[pbx-hlh-8p>nls::mCherry]* control larvae at the L4 stage. Mid-sagittal DIC images overlaid with the NLS::mCherry or LIN-39::mCherry signals (magenta) are shown, with the separate and merged fluorescence signals below

(SM > GFP, green). Panels show maximum intensity z-projections of the fluorescent signals in the left and right body halves, respectively. The 16 SM nuclei are indicated with black and white arrowheads and the extra SM > GFP, LIN-39::mCherry double-positive cells with yellow arrowheads. **(B')** Quantification of SM numbers in L4 larvae. Bars show the average numbers of SMs, error bars the standard deviation and dots the individual values. Two independent lines each for *pbx-hlh-8p>lin-39::mCherry* (*zhEx678.1 & 672.2*) and *pbx-hlh-8p>nls::mCherry* (*zhEx690.1 & 690.2*) transgenes were scored. The numbers of animals scored are indicated in brackets. Statistical significance was determined with a two-tailed Mann-Whitney non-parametric test and is indicated with ** for *p* = 0,0017. Scale bars are 20 μm. Source data are provided as a Source Data file.

expressing the *SM>gfp* reporter and *nls::mCherry* were detected (*n* = 41), except for two cases, where only eight *SM>gfp*-positive cells were found on one side, possibly because the animals were at a later developmental stage when *hlh-8* expression had already ceased (Fig. 3B, B'). In contrast, *pbx-hlh-8p>lin-39::mCherry* animals contained on average 21.7 ± 10.5 SD (*n* = 42) *SM>gfp*-positive cells, with five larvae containing over 32 SMs (Fig. 3B, B'). We noticed that many of the excess *SM>gfp* and *lin-39::mCherry* double-positive cells were irregularly distributed throughout the posterior body region (Fig. 3B). SMs over-expressing *lin-39* may have divided before reaching the midbody region, causing them to prematurely stop migrating[35]. Alternatively, over-expression of *lin-39* may have partially transformed other M lineage cells into SMs, causing these cells to proliferate without migrating.

We conclude that the over-expression of LIN-39 is sufficient to induce extra cell divisions of the SMs or their descendants.

### Ectopic *lin-39* expression in the anchor cell induces proliferation

To study the effects of ectopic *lin-39* expression, we chose the gonadal anchor cell (AC), as it did not show endogenous LIN-39::ZF1::GFP expression (Fig. 1A). Once specified during the early L2 stage, the AC remains arrested in the G1 phase of the cell cycle, invades the vulval epithelium during late L3 and fuses with neighboring VU cells during mid-L4[38,39].

Analogous to the SMs, we expressed a *lin-39::gfp* mini-gene under control of an AC-specific promoter/enhancer (ACEL)[40] with the 161 bp distal *egl-17* enhancer fragment inserted at the 5' end (*pbx-ACELp>lin-39::gfp*, Fig. 4A). As a negative control, a *pbx-ACELp>nls::gfp* transgene was used (Fig. 4A). Both transgenes were expressed from extra-chromosomal arrays (*zhEx650[pbx-ACELp>lin-39::gfp]* & *zhEx666[pbx-ACELp>nls::gfp]*) and combined with the *qyIs23[cdh-3p > PH::mCherry]* reporter (*AC>mCherry*), outlining the AC plasma membrane and the *qyIs10[lam-1::gfp]* reporter labeling the basement membranes (BMs) that separate the uterus from the vulva[41]. Over-expression of *lin-39::gfp* resulted in the formation of multiple LIN-39::GFP expressing cells (Fig. 4B). Forty-five percent of *pbx-ACELp>lin-39::gfp* larvae (*n* = 102) contained more than one AC, with an average of 4.9 ± 4.1 SD (*n* = 46) and up to 18 LIN-39::GFP-positive cells in affected animals (Fig. 4B' and Supplementary Fig. 4B). Furthermore, the proliferating ACs did not

breach the BMs. Even in *pbx-ACELp>lin-39::gfp* animals containing a single AC, BM breaching did not occur in 31% of the cases (*n* = 39).

To examine the timing of AC proliferation and also exclude the possibility that the excess LIN-39::GFP expressing cells were VU cells ectopically expressing the transgene rather than proliferating ACs, we performed live-imaging of animals carrying a chromosomally integrated version of the *pbx-ACELp>lin-39::gfp* transgene (*zhIs150*). The *zh150* transgene caused a similar AC multiplication phenotype as the extra-chromosomal array (Supplementary Fig. 5A). Observation of *pbx-ACELp>lin-39::gfp* animals from the late L2 stage until young adulthood revealed that the AC expressed LIN-39::GFP at relatively low levels during L2. The AC started to divide during the L3 and L4 stages in 63% of the cases (*n* = 30) when LIN-39::GFP expression levels dynamically increased and decreased (Supplementary Fig. 5D and Supplementary Movie S4). Expression faded during late L4 and disappeared in 74% (*n* = 50) of one-day-old adults (Supplementary Fig. 5B). It thus appears that the AC only started to proliferate once a certain threshold concentration of LIN-39::GFP has been reached. Since the *pbx-ACELp>lin-39::gfp* transgene was already expressed in the newly formed AC in early L2 larvae, it is possible that LIN-39 overexpression may have prevented the AC from entering a quiescent state during L2, but expression levels were only sufficiently high to induce proliferation later during L3.

To assess the fate of the proliferating ACs, we scored co-expression of the *AC>mCherry* marker driven by the AC-specific *cdh-3* enhancer element[41] in LIN-39::GFP-positive cells. In animals with up to eight LIN-39::GFP-positive cells, most of these cells also expressed *AC>mCherry*, but in animals with more than eight LIN-39::GFP-positive cells, most or all of these cells had ceased *AC>mCherry* expression (Fig. 4B and Supplementary Fig. 4A). To further examine the fate of the proliferating ACs, we scored the expression of a *lag-2p>gfp* reporter (*qIs56*)[42] in late L3 and early L4 larvae. Consistent with the above-mentioned observation, all animals with a single AC (*n* = 32) showed co-expression of LIN-39::mCherry and *lag-2p* > GFP, and 72% of animals with multiple ACs (*n* = 16) showed co-expression of the two markers (Supplementary Fig. 4C, C'). We conclude that in animals with up to eight ACs, the majority of the proliferating cells have kept an AC fate, but upon further proliferation, the cells may have lost their fate.

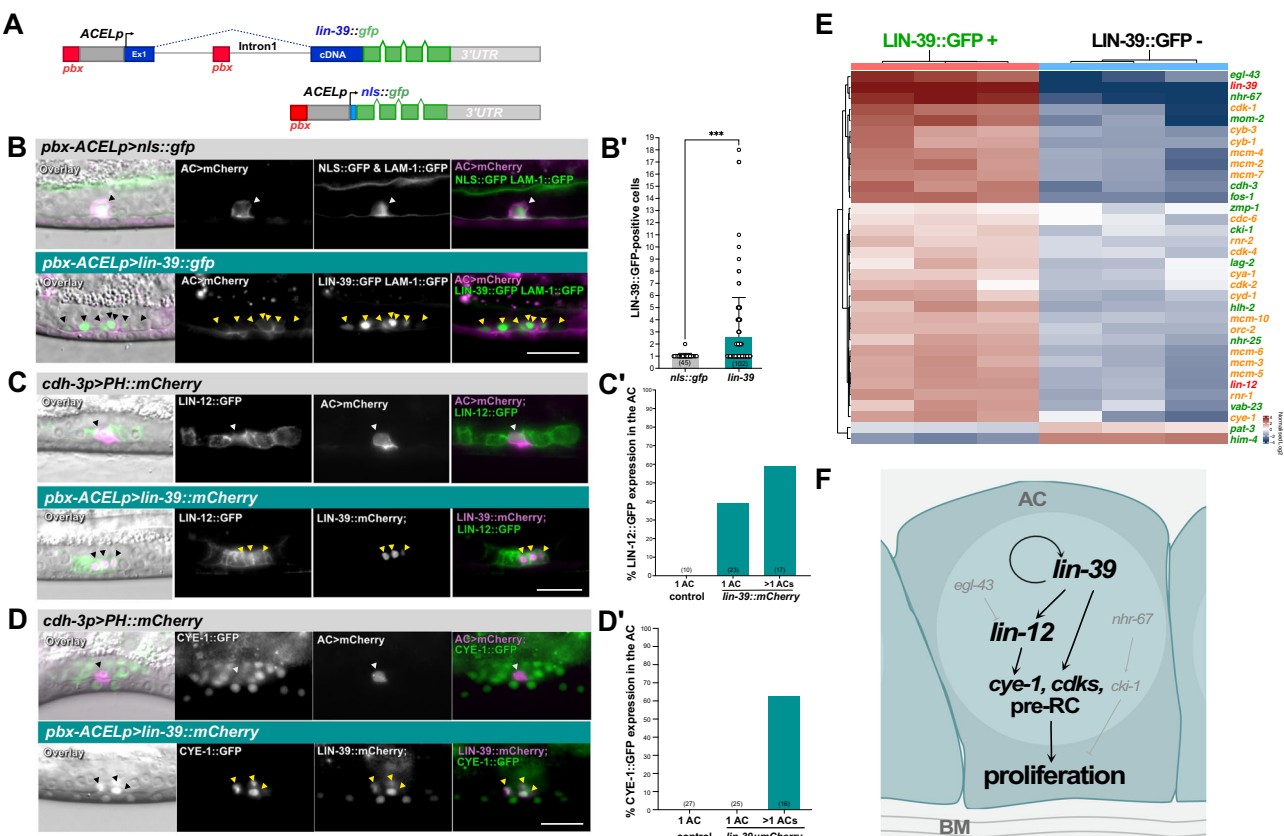

**Fig. 4 | Ectopic *lin-39 hox* expression causes AC proliferation. (A)** Schematic drawings of the transgenes used for AC expression. **(B)** AC proliferation in *zhEx666[pbx-ACELp>nls::gfp]* controls (top) versus *zhEx650[pbx-ACELp>lin-39::gfp]* (middle) and *zhEx660[pbx-ACELp>mab-5::gfp]* (bottom) L3 larvae. Mid-sagittal DIC images overlaid with the LAM-1::GFP (*qyIs10*) and NLS::GFP (top), LIN-39::GFP (middle) or MAB-5::GFP (bottom) (green) and AC>mCherry (magenta) signals are shown, along with the separate and merged fluorescence channels. **(B')** AC number in L3 to L4 larvae for the indicated transgenes. Bars show the average numbers of GFP-positive cells per animal, error bars the standard deviation and dots individual values. Two independent extra-chromosomal lines were analyzed for each transgene (*pbx-ACELp>nls::gfp*: *zhEx666.1 & 666.2*, *pbx-ACELp>lin-39::gfp*: *zhEx650.1 & 650.2*). Statistical significance was determined with a two-tailed Mann-Whitney non-parametric test and is indicated with *** for *p* = 3.2e-7. **(C)** LIN-12::GFP (*wgIs72*) expression in AC>mCherry control versus *zhEx689[pbx-ACELp>lin-39::mCherry]* L3 larvae. Mid-sagittal DIC images overlaid with the LIN-12::GFP (green) and AC>mCherry or LIN-39::mCherry (magenta) signals are shown, along with the separate and merged fluorescence channels. **(C')** Percentages of animals showing LIN-12::GFP expression in the AC of controls versus *pbx-ACELp>lin-39::mCherry* animals with one or multiple ACs. Two independent *pbx-ACELp>lin-39::mCherry* lines

(*zhEx689.1 & 689.2*) were scored. **(D)** CYE-1::GFP (*zhIs80*) expression in AC>mCherry control versus *zhEx689[pbx-ACELp>lin-39::mCherry]* L3 larvae. Mid-sagittal DIC images overlaid with the CYE-1::GFP (green) and AC>mCherry or LIN-39::mCherry (magenta) signals are shown, along with the separate and merged fluorescence channels. **(D')** Percentages of animals showing CYE-1::GFP expression in controls versus *pbx-ACELp>lin-39::mCherry*] animals with one or multiple ACs. Two independent *pbx-ACELp>lin-39::mCherry* lines (*zhEx689.1 & 689.2*) were scored. Single AC nuclei in (**B-D**) are labeled with black (DIC) and white arrowheads, multiple AC nuclei with yellow arrowheads. The numbers of animals scored are indicated in brackets. Scale bars are 20 μm. **(E)** Heat-map derived from the transcriptome analysis of isolated LIN-39::GFP-positive cells. Selected genes that were significantly up- or down-regulated relative to LIN-39::GFP-negative cells are shown. For the complete dataset of the three biological replicates including the statistical analysis, see Supplementary Data S1. Genes known to be expressed in the wild-type AC are labeled in green, genes encoding cell cycle and DNA replication regulators in orange, and genes normally not expressed in the AC in red. **(F)** The proposed mechanism, through which *lin-39* induces AC proliferation. Source data are provided as a Source Data file.

Given that LIN-39 is normally not expressed in the AC, we tested if lower levels of LIN-39::GFP might be sufficient to induce AC proliferation. To achieve this, we generated three transgenes delivering different dosages of *lin-39*; one transgene expressing *lin-39::gfp* under control of the ACEL promotor without the distal *egl-17* enhancer (*ACELp>lin-39::gfp*), a second containing a truncated, 50 bp distal *egl-17* enhancer (*short pbx-ACELp>lin-39::gfp*) and a third with the full-length 161 bp distal *egl-17p* enhancer re-inserted at the 5' end (*long pbx-ACELp>lin-39::gfp*) (Supplementary Fig. 4D-D''). In the absence of the distal *egl-17p* enhancer, no LIN-39::GFP expression and thus no AC proliferation could be observed (Supplementary Fig. 4D, D'', top). The addition of a truncated distal *egl-17* enhancer caused weak LIN-39::GFP expression but no AC proliferation (Supplementary Fig. 4D, D'', middle), while re-insertion of the full-length distal *egl-17* enhancer restored both LIN-39::GFP expression and AC proliferation (Supplementary Fig. 4D, D'', bottom). We

conclude that a positive-feedback loop maintaining high levels of LIN-39::GFP expression is necessary to induce AC proliferation.

Finally, we investigated if other *C. elegans hox* genes besides *lin-39* could induce AC proliferation. Expression of the anterior *hox* gene *ceh-13* under control of the same *pbx-ACELp* enhancer/promotor did not cause AC proliferation, but the expression of the central *mab-5* or posterior *egl-5 hox* gene[6] resulted in AC proliferation in 33% (*n* = 52) and 18% (*n* = 46) of the animals, respectively (Supplementary Fig. 6). Therefore, *mab-5* and *egl-5* may also be able to induce AC proliferation, either directly or indirectly through activation of endogenous *lin-39* expression.

### *lin-39* causes ectopic AC expression of *lin-12 notch* and *cye-1 cyclin E*

AC proliferation is controlled by a regulatory network formed by the zinc finger transcription factor *egl-43* and the nuclear hormone

receptor *nhr-67* that together maintain the AC arrested in the G1 phase of the cell cycle during invasion[43–45]. To investigate the mechanism, by which *lin-39* triggers AC proliferation, we investigated if depletion of *egl-43* or *nhr-67* affected the AC proliferation phenotype caused by *lin-39* over-expression.

RNAi-induced knock-down of *nhr-67*, but not of *egl-43*, enhanced AC proliferation in the *pbx-ACELp>lin-39::gfp* background (Supplementary Fig. 7A), suggesting that *lin-39* may act in parallel with *nhr-67* to induce AC proliferation. Since LIN-39 promotes LIN-12 NOTCH expression in the VPCs[46], we examined the expression of a *lin-12::gfp* reporter (*wgIs72*) in the AC of animals over-expressing *lin-39::mCherry* (*zhEx689[pbx-ACELp>lin-39::mCherry]*). In control animals carrying the *cdh-3p > PH::mCherry* AC marker, LIN-12::GFP expression was detected at the cortex of the VU cells but not in the AC (Fig. 4C, C'). In contrast, 39% (*n* = 23) of *pbx-ACELp>lin-39::mCherry* larvae with a single AC expressed LIN-12::GFP in the AC, and 59% (*n* = 17) of animals with more than one AC co-expressed LIN-12::GFP in the LIN-39::mCherry-positive cells (Fig. 4C, C'). To test if LIN-12 expression in the AC was necessary for LIN-39 to induce proliferation, we performed *lin-12* RNAi in *pbx-ACELp>lin-39::mCherry* animals. No significant suppression of the AC proliferation phenotype could be observed after *lin-12* RNAi (Supplementary Fig. 7A'), indicating that *lin-39* induces AC proliferation not exclusively by up-regulating *lin-12*.

Since *lin-39* directly activates *cye-1* expression in the VPCs[23], we examined if ectopic *lin-39* expression in the AC also induced *cye-1* expression. While *cye-1::gfp* was only faintly expressed in control animals and *pbx-ACELp>lin-39::mCherry* animals with a single AC, the majority of *pbx-ACELp>lin-39::mCherry* animals with multiple ACs exhibited strong CYE-1::GFP expression (Fig. 4D, D'). To test if the observed up-regulation of *cye-1* expression depends on *lin-12* activity, we examined the effect of *lin-12* RNAi on *cye-1::gfp* expression. *lin-12* RNAi in *pbx-ACELp>lin-39::mCherry* animals did not cause a significant reduction in the number of animals expressing CYE-1::GFP in the proliferating ACs (Supplementary Fig. 7B, B'). Thus, *lin-39* over-expression appears to promote AC proliferation by up-regulating the expression of *lin-12, cye-1*, and possibly additional target genes.

### *lin-39* up-regulates multiple cell cycle regulators and the DNA pre-replication complex

To gain further insights into the effects of ectopic *lin-39* expression in the AC, we extracted proliferating ACs from *pbx-ACELp>lin-39::gfp*; *mcm-4::mCherry* L3 larvae using the protocol developed by Zhang et al.[47]. GFP-positive cells were isolated using fluorescent activated cell sorting (FACS), achieving an enrichment of 80–90% GFP-positive cells (see Materials and Methods and Supplementary Fig. 8). Three independent isolates were used for transcriptome analysis by RNAseq. It should be noted that not all cell types are equally represented in cell suspensions extracted from *C. elegans* larvae, with muscle cells constituting the majority of isolated cells, though neuronal and uterine cells were also observed[47]. Since it was not possible to isolate a sufficient number of similarly enriched non-proliferating ACs from *pbx-ACELp>nls::gfp* control animals, we compared the transcriptome of LIN-39::GFP-positive (LIN-39::GFP + ) to LIN-39::GFP- & MCM-4::mCherry-negative (LIN-39::GFP-) cells extracted from *pbx-ACELp>lin-39::gfp* animals (Supplementary Fig. 8). Transcriptome analysis indicated a 395-fold enrichment of *lin-39* mRNA in LIN-39::GFP+ relative to LIN-39::GFP- cells (average of three biological replicates) (Fig. 4E and Supplementary Data S1). Moreover, mRNAs encoding the EGL-43, NHR-67, FOS-1 and HLH-2 transcription factors that are necessary to determine the AC fate and its invasive capacity were also enriched in LIN-39::GFP+ cells (Fig. 4E and Supplementary Data S1). Hence, the transcriptome analysis confirmed our conclusion from the *lag-2p>gfp* and *AC>mCherry* reporter analysis (Fig. 4B and Supplementary Fig. 4C, C'), which indicated that ectopic *lin-39* expression did not result in a complete re-programming of the AC fate. Furthermore, these results

indicated that ectopic *lin-39* expression could overcome the G1 cell cycle arrest, established by EGL-43 and NHR-67 in the AC[44,45]. Consistent with the *lin-12* and *cye-1* reporter analysis (Fig. 4C, D), mRNAs encoding these two genes were also enriched in LIN-39::GFP+ cells (Fig. 4E and Supplementary Data S1). In addition to these two known target genes, *lin-39* over-expression caused an up-regulation of many genes encoding core components of the cell cycle machinery, such as the G1 and G2/M-phase *cdk* and *cyclin* genes (*cdk-2, cdk-4, cyd-1, cyb-1* and *cyb-3*), as well as components of the pre-RC (*cdc-6* and the *mcm* genes) that are normally expressed at very low levels in the G1-arrested AC (Fig. 4E and Supplementary Data S1)[48].

Despite the persisting expression of the AC-specific transcription factors, expression of a few genes promoting AC invasion, such as the hemicentin *him-4* or the integrin β subunit *pat-3*, was decreased in LIN-39::GFP+ cells (bottom two rows in Fig. 4E). This could explain why the ACs in *pbx-ACELp>lin-39::gfp* larvae often failed to breach the BMs.

Taken together, the transcriptome analysis of proliferating ACs indicates that *lin-39* may promote AC proliferation by, directly or indirectly, up-regulating the expression of core cell cycle regulators and DNA replication factors (Fig. 4F).

### Preventing AC fusion extends the proliferative phase until young adulthood

In one of the *pbx-ACELp>lin-39::gfp* larvae observed by live-imaging, as mentioned above, the AC remained quiescent during L3 and L4, but divided once during adulthood (Fig. 5A and Supplementary Movie S5). Since somatic cells do normally not divide in adult *C. elegans*, we further characterized this phenomenon by live-imaging adult animals for longer periods. Most *pbx-ACELp>lin-39::gfp* animals had lost all LIN-39::GFP-positive cells after reaching adulthood (Supplementary Fig. 5B). We hypothesized that the fusion of the AC with adjacent VU cells forming the uterine seam cell (utse) syncytium during the L4 stage could be a cause of the disappearance of LIN-39::GFP-positive cells in adults. We therefore introduced the *aff-1(tm2214)* mutation into the *pbx-ACELp>lin-39::gfp* background to block AC fusion[49]. Since *pbx-ACELp>lin-39::gfp* animals are egg-laying defective, causing the larvae to hatch inside the uterus and obscure the view, we simultaneously inhibited fertilization by including the *fem-2(b245ts)* allele[50]. *aff-1(lf); fem-2(ts); pbx-ACELp>lin-39::gfp* animals were grown at 25 °C from the L1 stage onwards to block spermatogenesis and thereby extend the time, during which adults could be imaged.

Inhibiting AC fusion in *aff-1(lf); fem-2(ts); pbx-ACELp>lin-39::gfp* animals resulted in an increased frequency of adult animals retaining LIN-39::GFP-positive cells compared to *fem-2(ts); pbx-ACELp>lin-39::gfp* double or *pbx-ACELp>lin-39::gfp* single mutants (Supplementary Fig. 5B, C). Since the increase was more pronounced in animals grown at 20 °C, we raised *aff-1(lf); fem-2(ts); pbx-ACELp>lin-39::gfp* animals at 25 °C from the L1 until the late L4 stage and then down-shifted them to 20 °C. One-day-old adults (24 hours post L4) were immobilized in long-term microfluidic devices[26] and imaged at 20 °C for up to 72 hours. In three out of seven animals, the AC divided during the first 24 hours of imaging. In two out of these cases one round, and in one case two rounds of AC divisions could be observed (Fig. 5B and Supplementary Movie S6).

Since *aff-1(lf); fem-2(ts); pbx-ACELp>lin-39::gfp* adults are fragile and often did not survive prolonged live-imaging, we assessed the potential of LIN-39::GFP expressing cells to proliferate at later stages of adulthood by observing the *mcm-4::mCherry* reporter, which is normally not expressed in adult somatic cells[33]. One day-old adult control animals carrying a chromosomally integrated version of the *pbx-ACELp>nls::gfp* control transgene (*zhIs165*) contained a single NLS::GFP-positive cell in 4% (*n* = 50) of the cases, while 26% (*n* = 50) of *pbx-ACELp>lin-39::gfp* adults contained at least one LIN-39::GFP-positive cell (Supplementary Fig. 5B). MCM-4::mCherry was faintly expressed in the single AC of *pbx-ACELp>nls::gfp* control animals at the

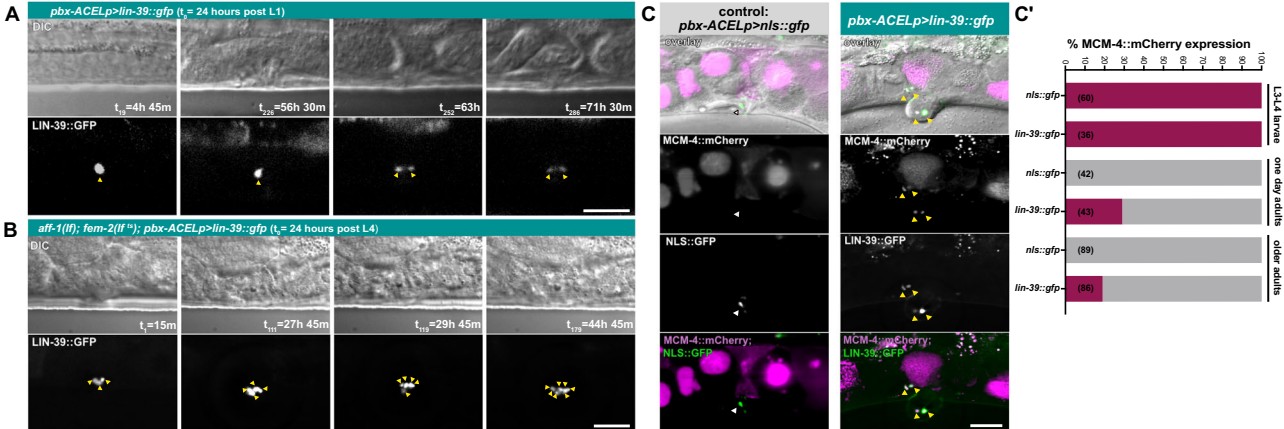

**Fig. 5 | Preventing cell fusion permits AC proliferation in adults.** Time-lapse image sequence of AC proliferation in a *zhIs150[pbx-ACELp>lin-39::gfp]* animal from the mid-L2 stage (24 hours post L1 starvation arrest) until young adulthood. (**B**) Time-lapse image sequence of AC proliferation in an adult *zhIs150[pbx-ACELp>lin-39::gfp] aff-1(tm2214); fem-2(b245ts)* animal. The recording started 24 hours post-L4. Images were taken at 15-minute intervals. (**A**) and (**B**) each show mid-sagittal DIC sections (top) and LIN-39::GFP maximum intensity projections (below). Time is indicated relative to the start of imaging $t_0$. Yellow arrowheads indicate AC nuclei. See Supplementary Movies S5 & S6 for all time points. (**C**) Expression of MCM-4::mCherry (*heSi45*) in an adult control *zhIs165[pbx-ACELp>nls::gfp]* (left) and a *zhIs150[pbx-ACELp>lin-39::gfp]* animal (right) approx. 55 hours post L4 stage. The top row shows mid-sagittal DIC images overlaid with maximum intensity projection of the NLS::GFP or LIN-39::GFP signals (green) and the MCM-4::mCherry signal (magenta). Individual and merged fluorescence channels are shown below. The white arrowhead indicates a single nucleus expressing NLS::GFP in the *zhIs165* control and the yellow arrowheads the nuclei co-expressing LIN-39::GFP and MCM-4::mCherry in *zhIs150*. (**C'**) Percentage of animals with MCM-4::mCherry-positive cells at the indicated stages. All MCM-4::mCherry-positive cells in the soma of *zhIs150* adults co-expressed LIN-39::GFP. Adult animals were scored 24 hours post-L4 (one-day adults) or 48 to 72 hours post-L4 (older adults). The numbers of animals scored for each condition are indicated in brackets. Scale bars are 20 μm. Source data are provided as a Source Data file.

early L4 stage (Supplementary Fig. 9), but absent in all adults, including the rare cases containing a single NLS::GFP-positive cell (Fig. 5C, C'). *pbx-ACELp>lin-39::gfp* animals, on the other hand, displayed strong MCM-4::mCherry expression in the LIN-39::GFP-positive ACs at the L4 stage (Supplementary Fig. 9). Moreover, MCM-4::mCherry; LIN-39::GFP double-positive cells were observed in 29% (*n* = 45) of one-day-old *pbx-ACELp>lin-39::gfp* adults, and their frequency decreased to 19% (*n* = 86) in two-day-old or older animals (Fig. 5C, C').

Thus, constitutive overexpression of *lin-39* can extend the capacity of the AC to proliferate until young adulthood.

## Discussion

Our results indicate that *hox* gene expression levels are a limiting factor for the proliferation of different somatic cell types in *C. elegans*. The *hox* gene *lin-39* is not only necessary to induce VPC cell cycle entry, as previously reported[22,23], but decreasing LIN-39 levels after vulval fate specification also causes premature cell cycle exit of the dividing vulval cells. Unexpectedly, constitutive over-expression of *lin-39* was sufficient to induce premature cell cycle entry of the Pn.p cells at the early L2 stage (Fig. 2A) and extend the proliferative phase of the vulval cells into late L4 (Fig. 2B, C). A similar over-proliferation phenotype was caused by *lin-39* over-expression in the SMs (Fig. 3), where *lin-39* and its co-factor *ceh-20* were previously found to activate the expression of the Sox C transcription factor *sem-2* to promote SM proliferation[35]. SMs over-expressing *sem-2* or *lin-39* were often mislocalized in the posterior body region, suggesting that both transcription factors induce premature division of the migrating SMs before they have reached the mid-body region[35]. The capacity of *lin-39* to induce proliferation was not limited to cell types that depend on *lin-39* activity during normal development. The gonadal AC did not exhibit any detectable expression of the endogenous LIN-39::ZF1::GFP reporter, yet ectopic *lin-39* expression caused the AC to re-enter the cell cycle and undergo multiple rounds of cell divisions (Fig. 4).

In all these cases, over-proliferation required a positive feedback loop created by the insertion of extra *hox/pbx* binding sites into the *lin-39* over-expression constructs. This positive feedback loop ensured that the initially low *lin-39* expression levels were amplified and

maintained at high levels beyond the normal period of cell proliferation, persisting until adulthood. Hence, constitutive *lin-39* over-expression permitted the vulval cells to undergo extra divisions during late L4 and the AC to divide during young adulthood. The prolonged proliferative capacity of somatic cells over-expressing *lin-39* also manifested in the persisting expression of the pre-RC gene *mcm-4* in young adults.

Ectopic AC expression of the second central *hox* gene *mab-5* and, to a lesser extent, the posterior *hox* gene *egl-5* likewise triggered AC proliferation. These *hox* genes may share a common activity that promotes cell division, as already shown for *lin-39* and *mab-5* in the M-lineage[35].

A possible mechanism, through which *lin-39* induces the proliferation of somatic cells, can be inferred from the transcriptome of proliferating, *lin-39*-expressing ACs. The observation that transcription factors, which are essential to specify the invasive AC fate and maintain the G1 cell cycle arrest[44,45], continued to be expressed in the dividing ACs suggests that *lin-39* over-expression did not induce proliferation by re-programming the AC fate. *lin-39* rather caused a broad activation of the cell cycle machinery to overcome the G1-arrest of the AC (Fig. 4F). Furthermore, many components of the preRC, which is necessary to license DNA replication origins in proliferating cells and, at lower levels, promote AC invasion[48], were highly enriched in *lin-39*-positive ACs. This may be of particular relevance because HOX proteins were found to be enriched at DNA replication origins, where they stimulate the assembly of the pre-RC in a transcription-independent manner[14,51]. Besides the cell cycle machinery and pre-RC components, *lin-39* also caused an up-regulation of the *lin-12 notch* receptor, while the *Delta* ligand *lag-2* continued to be expressed in the proliferating ACs. Hyper-activation of *lin-12* can rescue the proliferation defect of VPCs lacking *lin-39*[23], and it is sufficient to induce AC proliferation[44]. Though, LIN-39 does not appear to act exclusively via LIN-12 since *lin-39* caused *cye-1* up-regulation and AC proliferation, even when *lin-12* was down-regulated.

Our results are consistent with numerous reports showing that *hox* gene expression levels in different animals correlate with cell proliferation during embryogenesis, organ regeneration and tumor formation[14,51–53]. A connection between *hox* gene expression levels and

regeneration has first been observed in freshwater planarians[54], where *hox* gene expression is maintained at low levels in adult animals, but dramatically increases during regeneration and again decreases, once proliferation in the regenerating tissue has ceased. Also in mammals, expression of region-specific *hox* genes increases during the regeneration of adult organs[7–9]. Moreover, HOX protein levels vary during cell cycle progression and decrease in serum-starved or terminally differentiated mammalian cells[6,14,55]. The down-regulation of *hox* gene expression in quiescent cells may not only occur at the transcriptional level but also involve ubiquitin-mediated HOX protein degradation[56,57]. Cell cycle-dependent HOX protein degradation may explain why we observed fluctuating levels of LIN-39::GFP expression in the time-lapse recordings of proliferating ACs.

The role of *hox* genes in cell cycle regulation is of particular importance during cancer formation, where increased *hox* gene expression induced by oncogenes, growth factors or epigenetic changes drives tumor cell proliferation (reviewed in[6,7,12,13]). For example, ectopic HOXA1 expression is sufficient to cause the oncogenic transformation of mammary gland cells[58], over-expression of HOXA9 is required for the proliferation of myeloid leukemia cells[59], and HOXB13 promotes the growth of ovarian carcinoma[60]. The ability to stimulate cell proliferation thus appears to be a common function of *hox* genes.

We thus propose that the dosage of LIN-39 and other HOX proteins sets a threshold determining the temporal competence of somatic cells to proliferate during *C. elegans* larval development by maintaining the cell cycle in an active state. The down-regulation of *hox* gene expression in the majority of somatic cells towards the end of larval development may explain why adult somatic cells irreversibly exit the cell cycle and the soma cannot regenerate.

To our knowledge, this is the first study showing that constitutive expression of *C. elegans hox* genes can bypass the cell cycle arrest imposed by the fixed cell lineage and prolong the proliferative capacity of somatic cells until adulthood. A dramatic over-proliferation of the M-lineage has previously been achieved by the simultaneous down-regulation of the SWI/SNF complex and multiple cell cycle inhibitors[4]. In light of these findings, it is surprising that the over-expression of a single *hox* transcription factor without any of its co-factors is sufficient to induce a similar phenotype. It is therefore possible that *hox* genes are crucial downstream targets of the SWI/SNF complex during proliferation control.

In conclusion, our findings establish a direct, causal relationship between *hox* gene expression levels and cell cycle progression, suggesting that persisting *hox* gene expression during adulthood is necessary to maintain the regenerative capacity of somatic cells.

## Methods

### *C. elegans* maintenance
*C. elegans* were grown on standard Nematode Growth Medium (NGM) agar plates seeded with OP50 *E. coli* bacteria[61]. Animals were grown at 20 °C or 25 °C where indicated. The reference wild-type (*wt*) strain was *C. elegans* Bristol, variety N2. A list of strains used can be found in Supplementary Data S2.

### Generation of the *lin-39(zh120[lin-39::zf1::gfp])* allele
An in-frame insertion of a *zf1::gfp* tag after the last exon of the *lin-39* locus was obtained via CRISPR/Cas9-mediated homologous recombination as described[62]. The backbone for the donor template plasmid containing the Self-Excising Selection Cassette was amplified from pDD282 and the *lin-39* homology arms from genomic N2 DNA. Primers used for the generation of the sgRNA and donor-template plasmids are listed in Supplementary Data S3 and plasmids in Supplementary Data S4.

### Generation of transgenic lines
Plasmids were co-injected into the syncytial gonads of young adults at a concentration of 50 ng/μl with 50 ng/μl of pBS KS (-) and 2.5 ng/μl

ofpCFJ90 (*myo-2::mCherry*) co-injection marker[63]. Two to three independent transgenic lines were selected for each construct. The *zhIs150* integrated array was generated by UV irradiation of *zhEx650[pbx-ACELp>lin-39::gfp]* animals, *zhIs167* was obtained from *zhEx630[egl-17p>lin-39::gfp]* and *zhIs165* from *zhEx666[pbx-ACELp>nls::gfp]* by gamma-ray irradiation[64]. Integrated lines were outcrossed three times.

### RNA interference
Embryos were isolated by hypochlorite treatment of gravid adults and allowed to hatch overnight in M9 buffer to obtain synchronized L1 larvae that were plated on NGM plates containing 3 mM IPTG seeded with *E. coli* producing dsRNA[65]. RNAi clones were obtained from a *C. elegans* genome-wide or the *C. elegans* open reading frame (ORFeome) RNAi library (both from Source BioScience). Bacteria were grown in 2 mL of 2xTY medium, containing 200 μg/mL ampicillin and 25 μg/mL tetracycline at 37 °C and directly seeded on NGM plates containing 3 mM IPTG. As a negative control, the empty L4440 vector was used (labeled *ev* in all figures). All RNAi experiments were repeated twice.

### Conditional *lin-39* knock-down
For the ZIF-1-mediated conditional knock-down[27], a plasmid encoding *hsp-16>zif-1* (pSH12, Supplementary Data S4) was injected into the *lin-39(zh120); swIs79[ajm-1::gfp]; heSi45[mcm-4::mCherry]* strain to generate two independent extra-chromosomal lines (*zhEx616.1 & 616.2*). Synchronized L1 larvae were grown for 24 hours until mid-L2 when *zif-1* mediated knock-down was induced by heat shock at 31 °C for 50 minutes. Larvae were then washed with M9 Buffer, transferred either to *ev* control or *lin-39* RNAi plates and grown until L4.

### Microscopy and long-term imaging
DIC (Nomarski) and fluorescent images were acquired with a LEICA DM6000B microscope equipped with a Hammamatsu ORCA FLASH 4.0LT sCMOS camera and a 63x (N.A. 1.32) oil-immersion lens. z-stacks were acquired with a spacing of 0.2 to 0.4 μm.

Long-term imaging experiments were carried out as described previously[26], using L1-4 devices for animals from the mid/late L1 stage until the early L4 stage, L2-A for animals from the mid/late L2 stage until young adulthood, and L3-A devices for animals from the mid/late L3 stage until adulthood [this study]. L3-A devices were made following the design rules given in[26], increasing channel width, height and length (WxHxL, 30x25x900 μm) to accommodate animals further into adulthood. Image stacks were taken at 15-minute intervals with a Leica DMRA2 microscope equipped with an sCMOS camera (Prime BSI, Photometrics, USA) and a Spectra (Lumencor, USA) light source inside a custom-made incubator at 20 °C (Okolab, Italy). z-stacks of 0.5 μm were acquired using a piezo objective drive (MIPOS 100 SG, Piezosystems Jena, Germany). Images acquired in the microfluidic devices were first cropped to the region of interest using a custom-built MATLAB script, and fluorescent image stacks were deconvolved using the YacuDecu implementation of CUDA-based Richardson Lucy deconvolution in MATLAB. Images were analyzed using Fiji software[66].

### Pn.p cell duplication
To score Pn.p cell duplications, developmental stages were determined by measuring the gonad length in DIC images[67]. Animals were divided into three groups: early to mid-L2 stage (gonad 25-70 μm), late L2 stage (gonad 70-110 μm) and early to mid-L3 stage (gonad 110-150 μm). Pn.p and vulval cell numbers were scored in z-stack by counting cell nuclei labeled with the *VPC>mCherry* (*he317[eft-3p>LoxP-egl-13nls::bfp2-LoxP-egl-13nls::mCherry]; heSi220[lin-31p>cre]*[31]) reporter, the *zhIs167[egl-17p>lin-39::gfp]* or *zhEx670.1&2[egl-17p::nls]* transgenes and nuclei visible in the DIC images. AC proliferation was likewise scored in z-stacks by counting all cells expressing the *qyIs23[cdh-3p > PH::mCherry]*[41] and *pbx-ACELp>lin-39::gfp* markers (or *nls::gfp, ceh-13::gfp, mab-5::gfp, egl-5::gfp* in Fig. 4B).

## AC isolation and RNAseq analysis

For each of the three replicates, approximately 180'000 synchronized L1 larvae were grown for 23 to 25 hours at 25 °C until the mid-L3 stage on 10 cm NGM plates seeded with UV-irradiated E.coli OP50 to prevent bacterial re-growth. Larvae were collected, washed twice with water, and single-cell suspensions were obtained using the larval cell extraction protocol developed by Zhang et al. [47]. LIN-39::GFP-positive cells were isolated on a FACS Aria III 5 L cell sorter selecting for cells with a diameter of 3 to 5 μm and appropriate gates (Supplementary Fig. 8A, see the Source Data file for the analysis of the three repeats and the raw FACS data). In each experiment, between 20'000 and 30'000 LIN-39::GFP-positive and 50'000 LIN-39::GFP & MCM-4::mCherry-negative cells were isolated and either plated in L15 medium supplemented with 10% bovine fetal calf serum on collagen-coated glass slides to score the enrichment of LIN-39::GFP-positive cells (Supplementary Fig. 8B), or collected directly in RNA extraction buffer for transcriptome analysis. Total RNA was purified with the RNeasy Micro kit (Qiagen Cat. No. 74004) following the manufacturer's protocol, yielding around 1 to 3 ng of RNA per replicate. Quality control (Agilent Tape Station/Bioanalyzer), RNA library preparation and whole exome sequencing (Illumina Novaseq 6000 System) were performed in triplicates by the transcriptomics service of the Functional Genomics Center Zürich (FGCZ). Between 20 and 100 million reads per sample were obtained. Bioinformatic analysis by the FGCZ used an in-house analysis pipeline with the open-source tools STAR for read alignment and DESeq2 for differential gene expression analysis.

**Statistical analysis** was done with Prism software (GraphPad) using the non-parametric Mann-Whitney or Kruskal-Wallis tests with Dunn's correction for multiple comparisons, as indicated in the figure legends.

## Reporting summary

Further information on research design is available in the Nature Portfolio Reporting Summary linked to this article.

## Data availability

All data are included in the manuscript and the supplementary information. Source data are provided with this paper. The RNAseq data generated in this study have been deposited in the European Nucleotide Archive (ENA) at EMBL-EBI under accession code PRJEB65923. There are no restrictions on data availability. Source data are provided with this paper.

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

## Acknowledgements

We wish to thank all the members of the Hajnal laboratory for the numerous scientific discussions, the *Caenorhabditis Genetics Center*, which is funded by the NIH Office of Research Infrastructure Programs (P40 OD010440), the van den Heuvel laboratory for providing strains, the deMello laboratory for access to the microfluidics facility and for materials, and the University of Zurich Flow Cytometry facility for the cell sorting and the Functional Genomic Center Zurich for RNA sequencing data. This work was supported by a grant from the Swiss National Science Foundation no. 31003A-166580 to A.H., the Swiss Cancer Research Foundation no. 4377-02-2018 to A.H. and the University of Zurich Candoc (Forschungskredit) to S.H..

## Author contributions

Conceptualization: S.H., A.H.; Validation: S.E.; Investigation: S.H., S.B., S.E., M.D.; Formal analysis: S.H.; Writing – original draft: S.H.; Writing – review and editing: S.H., S.B., S.E., M.D., A.H.; Supervision: A.H.; Funding acquisition: S.H., A.H.

## Competing interests

The authors declare no competing interests.
