## [Peer Review File · Nature Communications]

REVIEWER COMMENTS

Reviewer #1 (Remarks to the Author):

The paper by Heinze et al. is a nice study to show that Lin-39 controls cell proliferation.

It shows in a very elegant manner that levels of a Hox transcription factor can directly control the proliferation state of a cell.

The work is certainly of high interest in the field. For a more general audience, the interest will be more limited. The authors could improve this perhaps by providing in the discussion more parallels to fly and vertebrate findings – even some links to cancer so the relevance becomes more apparent.

I have no major issues with the paper, I find it clearly written, though it may be a bit difficult for a non-*C. elegans* person to read through the many genotypes – unfortunately I do not have a better idea. Maybe Table S2 could add an extra column with key words, explaining what the different strains do, what is visible with these strains? (sort of a check list, when reading the paper)

While I have no major comments, I have a few minor comments:

Line 41” homeobox domain: This is not the correct nomenclature, and not used in the paper they cite (Gehring, Hiromi). homeobox refers to the DNA region, the homeobox encodes the homeodomain. The correct word is “homeodomain” or homeodomain protein. Please change this (the authors have also used the incorrect nomenclature in past publications).

Line 52 Reference for the *C. elegans* Hox complex is poor, not suitable. A more suitable one would be Aboobaker, A.A., and Blaxter, M.L. (2003). Hox Gene Loss during Dynamic Evolution of the Nematode Cluster. *Curr Biol* 13, 37–40. It also shows the evolution.

Line 84: Original reference to lin-39, I think it is appropriate for this paper to also cite the original seminal paper on lin-39 here, not sure why it has been omitted: Clark, S.G., Chisholm, A.D., and Horvitz, H.R. (1993). Control of cell fates in the central body region of *C. elegans* by the homeobox gene lin-39. *Cell* 74, 43–55.

Line 641: the *c. elegans* (typing error, space).

References: italics for “*C. elegans*” , (journal style?). Bibliographic software should be able to handle italics.

Discussion: Given the conserved nature of the Hox complex, it certainly would be of interest to the general readership to provide some more examples and literature references to other cases, where Hox genes were shown to be involved in cell proliferation or regulation of cell cycle genes, it seems that often this can be observed in the context of cancer. For example, one possible publication, Sánchez-Herrero E. Hox targets and cellular functions. *Scientifica* (Cairo). 2013, reviews a number of publications with putative targets of Hox genes. A cursory search reveals quite a few more. But it is not up to me to write a discussion paragraph for the authors here. My recommendation is to overhaul the discussion with a section on similarities to other species.

Reviewer #2 (Remarks to the Author):

This paper addresses the interesting question of whether decreased expression of hox genes is responsible for the endpoints of the fixed vulva lineages in *C. elegans*. It primarily focuses on *lin-39*, the Hox gene that has long been known for having many roles in vulva development.

The paper begins with nice live imaging of endogenous LIN-39 tagged with a ZF1 degron and a fluorescent protein, which showed that LIN-39 expression faded in most of the terminal cells of the vulval lineage by the early L4 stage, and faded in the remainder by the end of the L4 stage. This led to the hypothesis that *lin-39* sustains proliferation and downregulation of its expression is the reason the lineage terminates. To test this hypothesis, they used simultaneous heat-shock induction of the ubiquitin ligase ZIF-1 in VPCs plus *lin-39i* (RNAi) and observed reduced divisions, consistent with the hypothesis but not in itself compelling.

More compelling evidence comes from findings from ectopic expression of LIN-39 in VPCs and other cell types. These observations, all indicating a role for Hox genes in regulating the cell cycle to control lineage endpoints, are interesting and provocative, and make this paper of general interest.

To achieve ectopic expression in VPCs, the authors used high copy number arrays made with a construct that contains an autoregulatory element that resulted in continuous expression of LIN-39::GFP in all VPCs and their descendants. They found that the VPCs and other Pn.p cells divided precociously in the L2 stage when normally they remain in G1. Another nice use of live imaging showed that the vulval lineage also continues past the point at which they would normally end. Similarly, they looked at another postembryonic blast cell type, the mesodermal SMs, and showed that LIN-39 normally disappears at the termini of that lineage, and when LIN-39 is overexpressed, extra divisions are also observed.

The study continues by looking at the effect of expressing LIN-39 in a quiescent differentiated cell, the anchor cell of the gonad (AC), which does not normally express LIN-39. To do this experiment they use high copy number extrachromosomal arrays formed with a construct that along with the LIN-39 autoregulatory element also uses an element, ACEL, which is expressed in precursors to the AC as well as in the AC. This point should be clarified: the fact that LIN-39 expression is likely to begin before the terminal differentiation of the AC may be important for understanding what is happening at the end point of lineages. (It does not detract from the main point of the study.)

The proliferating ACs continue to express AC markers and became the subject of further analysis. The authors determined the transcriptome of these LIN-39-expressing ACs and found broad activation of cell cycle regulators, including components of the pre-replication complex and of LIN-12 Notch, which this group previously showed causes AC proliferation (but see comment below). They also do a nice experiment in which they block fusion of the AC with the uterus so that observation into adulthood could be studied using long-term imaging, and find that proliferation continues in young adulthood, with a hint that this may diminish with age (see also comment below).

Apart from its interest to *C. elegans* developmental biologists, the authors put their work into a broader context by discussing how hox gene expression has been found to correlate with cell proliferation during development and regeneration. I therefore think the findings in this study will be of interest to a broader audience studying these problems in other systems.

Specific questions/comments

Line 93: Any hypothesis about why VulA cells are different?

Lines 104-119, Fig. 1: I was puzzled by the chosen experimental approach for the loss-of-function experiment, specifically why they chose to use the ZF1 degron rather than the AID degron, which in principle would allow more precise temporal control by simply administering auxin. Please comment on why the ZF1 approach was used instead.

The text wording doesn't clearly indicate that the control includes the effect of heat shock, although it is clear from the Fig. legend. I also think the statement " The remaining vulval cells that formed an invagination continued to express the AJM-1::GFP adherens junction marker, indicating that down-regulation of lin-39 after vulval induction did not cause a loss of cell polarity" should be toned down because the treatment is unlikely to cause a null condition.

Lines 174-176: Isn't the "overinduction" phenotype unexpected if the early division in the L2 was simply creating additional VPCs? Why wouldn't all of the extra VPCs just adopt the 3o uninduced fate? Please clarify or comment.

Lines 253ff: As pointed out above, ACEL begins to drive expression before the quiescent AC differentiates. To make the claim that ectopic expression occurs in the quiescent AC, a promoter that drives LIN-39 expression only after the AC fully differentiates would be necessary.

It may also be worth clarifying explicitly: Is the mcm-4 reporter expressed in the AC? Does it become expressed when LIN-39 is ectopically expressed there?

Line 271: "consistent with previous reports showing that the AC must remain arrested in the G1 phase for invasion": the authors need not do anything about this, but I just want to say that there is a Biorxiv post (doi: <https://doi.org/10.1101/2023.03.16.533034>) suggesting that the AC can be invasive even if not cell-cycle arrested.

Lines 306-311: The point is pushed a little too hard. Given potential cross-regulation of Hox genes, I wondered if forced expression of these other genes led to misexpression of endogenous lin-39. These incidental observations might thus be better placed as a supplemental figure to keep the ms. focused on lin-39.

Lines 312-330: These genes are brought into the story without context. Please introduce them better.

What is known about the individual knock-downs? How do they know that egl-43i is working with this negative result?

"Since EGL-43 represses the expression of lin-12 in the AC and ectopic activation of LIN-12 signaling is sufficient to cause AC proliferation [40], we analyzed the expression of a lin-12::gfp reporter (wgl572) in animals over-expressing lin-39::mCherry from an extra-chromosomal array (zhEx689[pbx-ACELp>lin-39::mCherry]) ... we performed lin-12 RNAi in pbx-ACELp>lin-39::mCherry animals. No suppression of the AC proliferation phenotype could be observed after lin-12 RNAi..."

Martinez et al. (doi: [10.1242/bio.059668](https://doi.org/10.1242/bio.059668)) recently challenged ref. 40 from the Hajnal group. It would be helpful for the field to discuss the disagreement and say if/how the new results reported in this study

bears on the challenge, which may also necessitate some critical evaluation of the Martinez et al. paper. If there is no room in the text, this could be considered in a supplemental figure or in the Methods.

line 588 van den Heuvel

Figures and legends

Line 766. What is "the L4.4 animal"?

Several figure panels: the tiny white arrowheads pointing to tiny white cells are hard to distinguish from one other without enormous magnification. Please consistently make the arrowheads a different color from what they're pointing at, as is done in some figure panels.

Fig. 4E: what does the gene color code mean? And the different font sizes?

Fig. 4F: Just show the AC; the rest of the cartoon is confusing. It looks like the VPCs don't divide, and it isn't clear whether the same process is going on in the other ACs.

To reviewer #1 (reviewer's comments in *italic*)

The paper by Heinze et al. is a nice study to show that Lin-39 controls cell proliferation. It shows in a very elegant manner that levels of a Hox transcription factor can directly control the proliferation state of a cell.

The work is certainly of high interest in the field. For a more general audience, the interest will be more limited. The authors could improve this perhaps by providing in the discussion more parallels to fly and vertebrate findings – even some links to cancer so the relevance becomes more apparent.

We have expanded the corresponding part of the discussion (**lines 475-494**) to better highlight the relevance of our work for a broader spectrum of readers by citing several examples of other species and of human tumors, where *hox* genes have been implicated in proliferation control.

I have no major issues with the paper, I find it clearly written, though it may be a bit difficult for a non-C. elegans person to read through the many genotypes – unfortunately I do not have a better idea. Maybe Table S2 could add an extra column with key words, explaining what the different strains do, what is visible with these strains? (sort of a check list, when reading the paper)

We have added a column to table S2 describing the use of the different strains and referring to the figures where each strain was used. We hope this helps the readers follow our experiments.

While I have no major comments, I have a few minor comments:

Line 41" homeobox domain: This is not the correct nomenclature, and not used in the paper they cite (Gehring, Hiromi). homeobox refers to the DNA region, the homeobox encodes the homeodomain. The correct word is "homeodomain" or homeodomain protein. Please change this (the authors have alas used the incorrect nomenclature in past publications).

We have corrected the nomenclature in the introduction and abstract. (**lines 15 &41**)

Line 52 Reference for the C. elegans Hox complex is poor, not suitable. A more suitable one would be Aboobaker, A.A., and Blaxter, M.L. (2003). Hox Gene Loss during Dynamic Evolution of the Nematode Cluster. Curr Biol 13, 37–40. It also shows the evolution.

We have added the suggested reference (17) but kept ref 6 because this review shows a good comparison of the nematode hox cluster with other species.

Line 84: Original reference to lin-39, I think it is appropriate for this paper to also cite the original seminal paper on lin-39 here, not sure why it has been omitted: Clark, S.G., Chisholm, A.D., and Horvitz, H.R. (1993). Control of cell fates in the central body region of C. elegans by the homeobox gene lin-39. Cell 74, 43–55.

Yes, this is a clear omission on our side and we should have cited the original Clark et al. (1993) paper here. This has now been corrected (ref 25)

Line 641: the c. elegans (typing error, space).

References: italics for "C. elegans" , (journal style?). Bibliographic software should be able to handle italics.

The italics have been corrected in all references (our bibliographic software seems to discard the italics after each reformatting!). The references are now in the required style for Nature Communications. **(lines 615ff)**

Discussion: Given the conserved nature of the Hox complex, it certainly would be of interest to the general readership to provide some more examples and literature references to other cases, where Hox genes were shown to be involved in cell proliferation or regulation of cell cycle genes, it seems that often this can be observed in the context of cancer. For example, one possible publication, Sánchez-Herrero E. Hox targets and cellular functions. Scientifica (Cairo). 2013, reviews a number of publications with putative targets of Hox genes. A cursory search reveals quite a few more. But it is not up to me to write a discussion paragraph for the authors here. My recommendation is the overhaul the discussion with a section on similarities to other species.

We have modified and expanded this part of the discussion to include more examples from different species showing a connection between *hox* gene expression and proliferation and provide additional references for the role of different *hox* genes in tumor formation in humans. Especially, the review articles cited as refs. 6&7 contain many additional references to the roles of *hox* genes in proliferation in development and cancer **(lines 472-491)**.

To reviewer #2 (reviewer's comments in *italic*)

Specific questions/comments

Line 93: Any hypothesis about why VulA cells are different?

LIN-39 is known to be required for toroid morphogenesis. It is possible that LIN-39 expression in VulA persists for a longer time because VulA is the last toroid to be formed (**lines 94-95**).

Lines 104-119, Fig. 1: I was puzzled by the chosen experimental approach for the loss-of-function experiment, specifically why they chose to use the ZF1 degron rather than the AID degron, which in principle would allow more precise temporal control by simply administering auxin. Please comment on why the ZF1 approach was used instead.

At the time we performed these experiments, the AID system had just been introduced in *C. elegans* and was not yet established in our group. Nevertheless, combined with RNAi, the ZF1 degron system proved very effective in reproducibly down-regulating LIN-39 protein.

The text wording doesn't clearly indicate that the control includes the effect of heat shock, although it is clear from the Fig. legend. I also think the statement "The remaining vulval cells that formed an invagination continued to express the AJM-1::GFP adherens junction marker, indicating that down-regulation of lin-39 after vulval induction did not cause a loss of cell polarity" should be toned down because the treatment is unlikely to cause a null condition.

The conditions for the control are now explicitly mentioned in the text (**line 111**) and figure legend (**line 786**). The reason we used the AJM-1 marker was to test if the vulval cells fused with hyp7 under the conditions used. It is correct, we cannot exclude that the partial *lin-39* depletion was not sufficient to cause a loss of cell polarity, and now mention this as one possible scenario (**lines 120-122**).

Lines 174-176: Isn't the "overinduction" phenotype unexpected if the early division in the L2 was simply creating additional VPCs? Why wouldn't all of the extra VPCs just adopt the 3o uninduced fate? Please clarify or comment.

Yes, the observation that extra Pn.p cells sometimes differentiated was unexpected. However, we only observed induction of proximal Pn.p cells, suggesting that the induction of these duplicated cells depended on the AC signal. We conclude that *lin-39* overexpression is not sufficient to cause differentiation, but rather renders the cells more sensitive to the inductive signals (**lines 183-186**).

Lines 253ff: As pointed out above, ACEL begins to drive expression before the quiescent AC differentiates. To make the claim that ectopic expression occurs in the quiescent AC, a promoter that drives LIN-39 expression only after the AC fully differentiates would be necessary.

This is an important point. Even though the AC only started to divide during L3 when sufficiently high levels of LIN-39 were produced (**lines 284-289**), it seems likely that LIN-39 expression during L2 prevented the AC from entering the permanently quiescent state it

enters in the wild-type. We have changed this section on **lines 290-293** to clarify this point. It is difficult to distinguish the permanent quiescence of the wild-type AC from the temporary arrest the AC enters in *lin-39* overexpressing animals during L2. We have attempted to express *lin-39* using the later AC enhancer/promoter from the *cdh-3* gene linked to a *pbx* site for auto-amplification, but for unknown reason this transgene only caused very faint AC expression and therefore did not cause AC proliferation.

It may also be worth clarifying explicitly: Is the mcm-4 reporter expressed in the AC? Does it become expressed when LIN-39 is ectopically expressed there?

The new **Figure S9** shows the weak MCM-4 expression level in the single AC of wild-type L4 larvae and the up-regulation in the multiple ACs of *lin-39* overexpressing L4 larvae. The graph shown in **Figure 5C'** now includes the frequency of MCM-4 expression at the L4 stage (without showing the differences in expression levels), but mention this point in the results section on **lines 417-419**.

Line 271: "consistent with previous reports showing that the AC must remain arrested in the G1 phase for invasion": the authors need not do anything about this, but I just want to say that there is a Biorxiv post (doi: <https://doi.org/10.1101/2023.03.16.533034>) suggesting that the AC can be invasive even if not cell-cycle arrested.

We have modified this point to leave room for exceptions (**lines 276-278**).

*Lines 306-311: The point is pushed a little too hard. Given potential cross-regulation of Hox genes, I wondered if forced expression of these other genes led to misexpression of endogenous *lin-39*. These incidental observations might thus be better placed as a supplemental figure to keep the ms. focused on *lin-39*.*

As suggested, we moved the data with the other *hox* genes to the new supplementary **Figure S6**. We also mention the possibility of cross-regulation, i.e. that other *hox* genes may act indirectly by up-regulating *lin-39* expression (**lines 318-323**).

Lines 312-330: These genes are brought into the story without context. Please introduce them better.

We have included a brief introduction of the two transcription factors that are known to inhibit AC proliferation and our rationale for examining the interaction with *lin-39* overexpression (**lines 325-329**).

*What is known about the individual knock-downs? How do they know that *egl-43i* is working with this negative result?*

egl-43i* must have been effective in our RNAi experiments (now shown in **Figure S7A) because we observed AC multiplication in either condition (*egl-43 i* with & without *lin-39* overexpression). Had *egl-43i* been ineffective, then we would have observed the same number of ACs as in the ev negative controls, which was not the case (p-values for *egl-43i* vs ev with & without *lin-39* overexpression are 0.002 and 0.01, respectively).*

*"Since EGL-43 represses the expression of *lin-12* in the AC and ectopic activation of LIN-12 signaling is sufficient to cause AC proliferation [40], we analyzed the expression of a *lin-12::gfp* reporter (*wgIs72*) in animals over-expressing *lin-39::mCherry* from an extra-chromosomal array (*zhEx689[pbx-ACELp>lin-39::mCherry]*) ... we performed *lin-12* RNAi in *pbx-ACELp>lin-**

39::mCherry animals. No suppression of the AC proliferation phenotype could be observed after *lin-12* RNAi...”

Martinez et al. (doi: 10.1242/bio.059668) recently challenged ref. 40 from the Hajnal group. It would be helpful for the field to discuss the disagreement and say if/how the new results reported in this study bears on the challenge, which may also necessitate some critical evaluation of the Martinez et al. paper. If there is no room in the text, this could be considered in a supplemental figure or in the Methods.

We are well aware of the publication by Martinez et al., but this is a completely unrelated issue that does not affect this manuscript and merits its own response. The data reported in this manuscript do not address this issue, but we have changed the corresponding paragraph (**lines 331-343**) to emphasize the regulation of *lin-12* by *lin-39*, and included a new reference, where regulation of *lin-12* by *lin-39* in the VPCs was first described (new ref 46). Since we observed LIN-12::GFP expression in the AC after *lin-39* over-expression, we wanted to test a possible involvement of *lin-12* in AC proliferation.

At the moment, we know that a *lin-12(lf)* mutation (*n137n720*) suppresses the AC proliferation phenotype caused by *egl-43* RNAi to a similar degree as *lin-12* RNAi in the experiment reported by Deng et al. (Ref 40). Hence, it seems unlikely that the double *egl-43 lin-12* RNAi experiment was ineffective. Possibly, the *lin-12* degron allele used by Martinez et al was not strong enough to suppress *egl-43i*.

line 588 van den Heuvel
corrected

Figures and legends

Line 766. What is "the L4.4 animal"?

We use the nomenclature introduced by Mok et al. (Ref.32) (L4.1 to L4.9) to distinguish the individual sub-stages of L4.

Several figure panels: the tiny white arrowheads pointing to tiny white cells are hard to distinguish from one other without enormous magnification. Please consistently make the arrowheads a different color from what they're pointing at, as is done in some figure panels.

We tried to use a different color coding for the arrowheads and took care to move the arrowheads out of the animal bodies wherever possible, but replacing the white arrowheads with colored ones created problems in the merged fluorescence images. We therefore think this is the best possible compromise.

Fig. 4E: what does the gene color code mean? And the different font sizes?

We now use the same font size for all gene names in **Figure 4E**. The color-coding is explained in the figure legend (**lines 868-870**) to distinguish between gens normally expressed in the AC (green), cell cycle and DNA replication regulators (orange) and genes that are not expressed in the wild-type (red).

Fig. 4F: Just show the AC; the rest of the cartoon is confusing. It looks like the VPCs don't divide, and it isn't clear whether the same process is going on in the other ACs.

Figure 4F was changed as suggested.

The FACS data with the gates used for cell sorting are now shown in **Fig. S8** for one of the three experiments, and the **suppl. Data S1** contains the analysis of all three experiments used for the RNAseq analysis, including the raw FACS data files.

REVIEWERS' COMMENTS

Reviewer #1 (Remarks to the Author):

The authors have amended their manuscript in a satisfactory manner, and it can be published in my opinion.

Reviewer #2 (Remarks to the Author):

I liked the paper originally, and I am pleased with the response to my comments. Congratulations to the authors on such a nice study!